# CROSS-LINGUAL TRANSFER WITH LARGE LANGUAGE MODELS VIA ADAPTIVE ADAPTER MERGING

## ABSTRACT

As an effective alternative to the direct fine-tuning on target tasks in specific languages, cross-lingual transfer addresses the challenges of limited training data by aligning representations across languages or by explicitly translating target languages into source languages. However, these methods possess certain limitations and fail to fully exploit the potential of Large Language Models (LLMs). In this paper, we regard the ability of LLMs in a particular task and language as a combination of "task ability" and "language ability". In the context of parameter-efficient fine-tuning and cross-lingual transfer, task ability can be obtained by adapters fine-tuning on the target task in the source language, while language ability is the ability to solve problems using the specific target language. In this work, we propose a novel adaptive adapter merging method for cross-lingual transfer, termed as `AdaMergeX`. As language ability is not tied to any specific task, we introduce another easily accessible reference task from which language ability is obtained by adapter merging. Then by further merging it with adapters tuned on the target task in the source language, we can achieve effective cross-lingual transfer. Furthermore, unlike existing model merging methods that employ arithmetic addition, we propose a new structure-adaptive merging method that adapts the merging process based on the structure of adapters. Our empirical results demonstrate that our approach yields new and effective cross-lingual transfer, outperforming existing methods across all settings.

## 1 INTRODUCTION

Multilingual NLP models, including conventional models such as mBERT (Kenton & Toutanova, 2019), XLM (Conneau & Lample, 2019), XLM-R (Conneau et al., 2020), as well as recent multilingual large language models (LLMs) like ChatGPT (OpenAI, 2022), PaLM2 (Anil et al., 2023), Llama2 (Touvron et al., 2023), have gained significant attention in response to the growing need for multilingual requirements. To further enhance the model's capability, particularly in cases where training data of certain tasks for low-resource languages is scarce and fine-tuning becomes impractical (Ma et al., 2023), cross-lingual transfer is introduced to extend the task-solving ability in a source language (e.g., English) to a wide range of languages. Various cross-lingual transfer techniques have been investigated, including extracting similar representations (Nguyen et al., 2023; Salesky et al., 2023; Gao et al., 2023) and translating to intermediate languages (Liang et al., 2023; Huang et al., 2023b). However, studies have indicated that aligned representations do not significantly benefit cross-lingual transfer, especially for LLMs (Gaschi et al., 2023). Additionally, the prompting method, specifically developed for LLMs, exhibits certain constraints when applied to low-resource languages due to the restricted optimization space, resulting in an incomplete exploration of the capabilities offered by LLMs (Li et al., 2023; Tanwar et al., 2023; Zhang et al., 2023b).

In this work, we regard the ability of LLMs in a particular task and language as a combination of "task ability" and "language ability". The former denotes the model's competence in performing a certain task, whereas the latter signifies their general proficiency of a certain language. Considering that language ability is not tied to any specific task, in line with the renowned equation "$king - queen = man - woman$", we perceive the disparity between the corresponding elements on each side of the equation as reflective of language proficiency. In the case of parameter-efficient fine-tuning (PEFT), we assume that the divergences between adapters fine-tuned in different languages on a particular

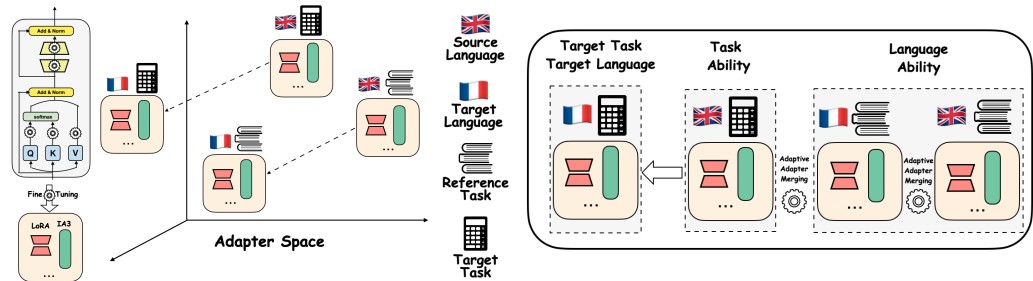

Figure 1: `AdaMergeX` split target task ability in the target language into "task ability" and "language ability". In the context of PEFT, "task ability" is obtained by fine-tuning on the target task in the source language. To achieve cross-lingual transfer, we introduce another reference task aimed at acquiring language ability. This reference task can be conveniently constructed using unlabeled data from both the source language and the target language. Moreover, by employing adaptive adapter merging, `AdaMergeX` combines the task ability and language ability to effectively adopt the adapter for the target task in the target language.

task follow the same distribution across diverse tasks. In the context of cross-lingual transfer, where labeled data is available only for the target task in the source language and there is unlabeled data in both the source and target languages, we aim to transfer the task ability from the source language to the target language. Specifically, task ability can be obtained by fine-tuning on the target task in the source language, while the language ability, as analyzed by the renowned equation, is quantified by the divergence between adapters of the task in the source language and the target language. Based on the assumption that the divergences between adapters fine-tuned in different languages for a particular task follow a consistent distribution across diverse tasks, we propose to accomplish cross-lingual transfer through Adaptive Adapter Merging (`AdaMergeX`) as shown in Figure 1. We introduce another reference task to obtain adapters that represent language ability. The reference task refers to an easily accessible task with unlabeled training corpus available for both high-resource and low-resource languages, such as causal language modeling. Generally speaking, we employ three types of adapters: those tuned on the target task in the source language, those tuned on the reference task in the target language, and those tuned on the reference task in the source language. The first type of adapters represents task ability, while the divergence between the last two types represents language ability. By merging these three types of adapters, we can obtain the adapter applicable to the target task in the target language.

Furthermore, in contrast to previous studies that combine models or adapters through a linear combination (Ilharco et al., 2022; Zhang et al., 2023a; Yadav et al., 2023; Huang et al., 2023a; Chronopoulou et al., 2023; Ponti et al., 2023), we argue that the model merging method should align with the manner in which adapters are integrated with language models. Specifically, for LoRA (Hu et al., 2021) that incorporates adapters through element-wise addition, the merging method should also involve element-wise addition or subtraction. Conversely, for $(IA)^3$ (Liu et al., 2022), which incorporates adapters through element-wise multiplication, the merging method should also employ element-wise multiplication or division. To the best of our knowledge, we are the first ones to apply adapter merging method to cross-lingual transfer, and we are the first one to design a structure-adaptive adapter merging method, which we find to be essential for the cross-lingual transfer problem.

We evaluate `AdaMergeX` on a total of 5 multilingual tasks spanning 12 languages, covering a broad resource spectrum from high-resource to low-resource language. Our evaluation demonstrates that `AdaMergeX` consistently outperforms other state-of-the-art methods both in cross-lingual transfer via prompting and general adapter merging methods. Notably, compared to XLT (Huang et al., 2023b), which is the state-of-the-art cross-lingual transfer method with LLMs, `AdaMerges` achieves $67.3\%$ and $11.3\%$ relative improvement on average in all languages and all tasks with and without fine-tuning on English labeled data respectively. In the case of state-of-the-art adapter merging method Arimerge (Zhang et al., 2023a), `AdaMergeX` achieves $31.1\%$ relative improvement on average in all languages and all tasks with Llama2. Moreover, the ablation analysis shows that `AdaMergeX` performs consistently well with different backbone models, different source languages, and different reference tasks.

## 2 BACKGROUND

Given a pre-trained model, fine-tuning is often employed to improve the performance on specific tasks. Specifically, for a layer $h = W_0x$, where $x \in \mathbb{R}^k$ is input, $h \in \mathbb{R}^d$ is output and $W_0 \in \mathbb{R}^{d \times k}$ is pre-trained parameters, fine-tuning updates parameters from $W_0$ to $W'$ and the layer becomes $h = W'x$. However, full fine-tuning requires many training data points and computing resources, which inspires the design of adapters (Houlsby et al., 2019).

With adapters, the layer is changed to $h = (W_0 \circ W_A)x$, where $W_A$ denotes the parameters of adapters and $\circ$ denotes the combination operation of pre-trained parameters and adapter parameters. During such parameter-efficient fine-tuning, pre-trained parameters $W_0$ are fixed and only adapter parameters $W_A$ are updated. Since the number of parameters of the adapter is often much fewer than the original model, adapters provide an effective way of fine-tuning on specific tasks. With the number of parameters growing much bigger for LLMs, adapters become more widely used in the current practice of fine-tuning LLMs (Hu et al., 2021; Li & Liang, 2021; Liu et al., 2022)

Various combination methods $\circ$ have been designed for different adapters. In this paper, we focus on two main widely used combination methods: addition and multiplication, corresponding to LoRA (Hu et al., 2021) and (IA)$^3$ (Liu et al., 2022), respectively:

**LoRA** Specializing the combination method "$\circ$" to element-wise addition denoted as "$\oplus$", LoRA employs low-rank decomposition to reduce training complexity. The layer is thus changed to

$$h = (W_0 \oplus W_A)x = (W_0 \oplus BA)x, \tag{1}$$

where $B \in \mathbb{R}^{d \times r}$ and $A \in \mathbb{R}^{r \times k}$ are low-rank decomposed matrices, and the rank $r \ll \min(d, k)$. Specifically, the LoRA can be implemented in any layer of the Transformer (Vaswani et al., 2017) architecture, including the attention layer and the feed-forward layer.

**(IA)$^3$** (IA)$^3$ specializes the combination method to element-wise multiplication "$\odot$":

$$h = (W_0 \odot W_A)x, \tag{2}$$

where $W_A \in \mathbb{R}^k$ is element-wise multiplied to each row of $W_0$. Furthermore, (IA)$^3$ can only be implemented to the key neuron and value neuron in the attention layer and dimension reduction neuron in the feed-forward layer.

## 3 ADAMERGEX: ADAPTIVE ADAPTER MERGING FOR X-LINGUAL TRANSFER

### 3.1 CROSS-LINGUAL TRANSFER VIA ADAPTER MERGING

Generally, the ability of a model in a particular task and language can be seen as a composite of two abilities, namely, "task ability" and "language ability". The former denotes the model's competence in performing a certain task (e.g., text classification, sentence completion), whereas the latter signifies their general proficiency in the given language (e.g., English, Chinese, German). By embracing the fundamental concept behind the renowned equation "$king - queen = man - woman$" and extending its applicability to linguistic contexts, we interpret the disparity between the respective terms on either side of the equation as indicative of language ability. In the case of parameter-efficient fine-tuning with LLMs, since language ability is not tied to specific tasks, we make the assumption that the divergences between adapters fine-tuned in different languages on a particular task follow the same distribution across diverse tasks.

Formally speaking, $A_{l_i t_j}$ denotes the adapter of task $t_j$ in language $l_i$, then for any two languages $l_1$, $l_2$ and two NLP tasks $t_1$, $t_2$, we have

$$A_{l_1 t_1} \| A_{l_2 t_1} \sim A_{l_1 t_2} \| A_{l_2 t_2}, \tag{3}$$

where $\|$ denotes the divergence among two adapters. For example, let's consider $l_1$ and $l_2$ as English and German, respectively, and $t_1$ and $t_2$ as the text classification task and question answering task, respectively. Assuming we have training data for each task in both languages, we can fine-tune LLMs to obtain four adapters: text classification in English, text classification in German, question answering in English, and question answering in German. We assume that the divergence between

adapters for the text classification task in English and German, as well as the divergence between adapters for the question answering task in English and German, follows the same distribution. This divergence represents the "language ability" that is independent of specific tasks.

In the context of cross-lingual transfer, we aim to solve the task $t_1$ for the target language $l_1$, with the knowledge transferred from a source language $l_2$, which is often a high-resource language such as English. By imposing the condition of cross-lingual transfer, where labeled data is available only for the target task in the source language and there is unlabeled data in both the source and target languages, we can introduce another "reference task" $t_2$. This task can be easily constructed using unlabeled data, and language ability can be obtained by $A_{l_1t_2}\|A_{l_2t_2}$. Moreover, to obtain the ability of performing target task $t_1$ in the target language $l_1$, we can further transform Equation (3) as:

$$A_{l_1t_1} = A_{l_2t_1} \|^R (A_{l_1t_2}\|A_{l_2t_2}),\tag{4}$$

where $\|^R$ is the reverse function of $\|$. Intuitively, $A_{l_2t_1}$ represents the "task ability" in the source language, while $A_{l_1t_2}\|A_{l_2t_2}$ represents the "language ability". Through merging these two terms, we can transfer the "task ability" of $t_1$ from $l_2$ to $l_1$.

To transfer the knowledge from labeled data in the high-resource language (i.e., given $A_{l_2t_1}$), the next step is to specify the reference task $t_2$. We observe that there are many easily obtained corpora of low-resource languages, such as Wikipedia, online blogs, etc. These corpora can be used to construct intuitive tasks such as causal language modeling, which can serve as the reference task $t_2$. Simultaneously, we can also construct such tasks for the high-resource language $l_2$. Therefore, adapters can be fine-tuned on such easily accessible reference tasks in different languages to obtain $A_{l_1t_2}$ and $A_{l_2t_2}$. Cross-lingual transfer thus can be achieved by merging these three adapters.

## 3.2 STRUCTURE-ADAPTIVE ADAPTER MERGING

As introduced in Section 2, adapters have different structures, which inspires us to devise different adapter merging methods accordingly. We propose that the adapter merging approach must align with the way that the adapter combined with the original model.

**LoRA** In the fine-tuning process of LoRA, where the method involves element-wise addition to the original parameters, the merging method used to combine task ability and language ability should also employ element-wise addition. Additionally, since the divergence calculation approach $\|$ is intended to be the inverse function of the merging method, it should be carried out through element-wise subtraction in this scenario. Therefore, Equation (3) is equivalently transferred to

$$A_{l_1t_1} \ominus A_{l_2t_1} \sim A_{l_1t_2} \ominus A_{l_2t_2},\tag{5}$$

where $\ominus$ denotes element-wise subtraction, and Equation (4) is equivalently transferred to

$$A_{l_1t_1} = A_{l_2t_1} \oplus t \cdot (A_{l_1t_2} \ominus A_{l_2t_2}),\tag{6}$$

where $\oplus$ denotes element-wise addition and $t$ is the hyper-parameter that adapts the scale of two distributions in the same family of distributions.

**(IA)$^3$** Similarly, the fine-tuning method of (IA)$^3$ is element-wise multiplication to the original parameters, and the merging method should also be element-wise multiplication. Furthermore, we need to employ element-wise division to obtain the divergence between $A_{l_1t_2}$ and $A_{l_2t_2}$. Therefore, Equation (3) is equivalently transferred to

$$A_{l_1t_1} \oslash A_{l_2t_1} \sim A_{l_1t_2} \oslash A_{l_2t_2},\tag{7}$$

where $\oslash$ denotes element-wise devision, and Equation (4) is equivalently transferred to

$$A_{l_1t_1} = A_{l_2t_1} \odot \Big( \big(t \cdot (A_{l_1t_2} \oslash A_{l_2t_2}) - \mathbb{1}\big) + \mathbb{1}\Big),\tag{8}$$

where $\odot$ denotes element-wise multiplication and $t$ is the hyper-parameter that determines the scale of two distributions in the same family of distributions.

Moreover, in the case of other adapter structures such as Adapter (Houlsby et al., 2019) and Prefix-Tuning (Li & Liang, 2021), which involve the insertion of layers and prefix tokens into the model, the

merging process necessitates transferring adapters within the same space, such as MLP. In this paper, we primarily focus on LoRA and (IA)³ due to the lack of training data in target language for AdapterH and the subpar performance of prefix-tuning on fine-tuning (He et al., 2021). However, in the case of smaller language models such as XLM-R (Conneau et al., 2020), we implement `AdaMergeX` on it. The adaptive merging method is $A_{l_1 t_1} = t \cdot (A_{l_1 t_2} * A_{l_2 t_2}^{-1}) * A_{l_2 t_1}$, where $*$ represents matrix multiplication and $A_{l_2 t_2}^{-1}$ represents Moore-Penrose pseudo-inverse of the matrix. The experiment results are shown in Appendix A.1.

### 3.3 ADAMERGEX

Following notations in Section 3.1, to solve a target task $t_1$ in a target language $l_1$, i.e., obtain the adapter $A_{l_1 t_1}$, we need to fine-tune another three adapters: adapters on the target task in the source language ($A_{l_2 t_1}$), adapters on the reference task in the target language ($A_{l_1 t_2}$), and adapters on the reference task in the source language ($A_{l_2 t_2}$). Note that $A_{l_1 t_2}$ and $A_{l_2 t_2}$ are easily obtainable, as we can choose any task in the target and source language. As mentioned earlier, the task can even be causal language modeling, which only requires randomly selected corpora. Therefore, with only unlabeled data in both source and target language, our proposed `AdaMergeX` effectively transfers the target task proficiency from the source language to the target language.

In the case of LoRA, which fine-tunes LLMs by tuning $\{B, A\}$ in tuned layers of LLMs as introduced in Equation (1), adapters are merged following Equation (6) by element-wise addition and subtraction on $\{B, A\}$ in the corresponding layers of $A_{l_2 t_1}$, $A_{l_1 t_2}$, and $A_{l_2 t_2}$. On the other hand, in the case of (IA)³, the fine-tuning parameters are $W_A$ in tuned layers as depicted in Equation (2). Thus the merging method follows Equation (8), which involves performing element-wise multiplication and division of the corresponding layers of $A_{l_2 t_1}$, $A_{l_1 t_2}$, and $A_{l_2 t_2}$.

## 4 EXPERIMENTS

### 4.1 EXPERIMENTAL SETUP

**Datasets and Language**    To evaluate the effectiveness of our method, we conduct experiments on a wide variety of multilingual tasks in three main categories: reasoning tasks, natural language understanding (NLU) tasks, and natural language generation (NLG) tasks. For reasoning tasks, we test on multilingual arithmetic reasoning dataset XGSM (Shi et al., 2022) and multilingual common-sense reasoning dataset XCOPA (Ponti et al., 2020). For NLU tasks, we test on the multilingual natural language inference dataset XNLI (Conneau et al., 2018), and question-answering dataset XQuAD (Artetxe et al., 2020). For NLG tasks, we test on multilingual summarization dataset XLSum (Hasan et al., 2021). We choose 12 languages that appear in more than once in the above datasets, including German (de), Russian (ru), French (fr), Spanish (es), Chinese (zh), Vietnamese (vi), Turkish (tr), Arabic (ar), Greek (el), Thai (th), Hindi (hi), and Swahili (sw). The order is presented by the percentage of the language in the whole corpus from Common Crawl Monthly Archives [1]. Detailed settings of the size of test set and zero-shot prompts are shown in Table 1. We utilize intuitive prompting methods for all tasks except for XCOPA and XNLI, where we employ prompts from Huang et al. (2023b). Detailed examples of the prompting approach can be found in Appendix A.2. For MGSM and XCOPA, we adopt the whole test set, while for XNLI, XLSum, and XQuAD we randomly sample 1000, 500, and 1000 data points from the whole test set respectively.

**Baselines**    We conducted a comparison between our proposed method, which utilizes model merging for achieving cross-lingual transfer, and five competing techniques. These techniques include: (i) Vanilla zero-shot prompting ("Vanilla"), which directly assesses target languages using the pre-trained LLM. (ii) English Tuning ("Eng-FT"), which involves fine-tuning the model in English for target tasks and subsequently transferring it directly to target languages. (iii) Cross-Lingual-Thought Prompting ("XLT (Vanilla)") (Huang et al., 2023b) achieves state-of-the-art results on cross-lingual transfer with LLMs through carefully designed prompt template, which involves explicit translation from the target to the source language, reasoning in the source language, and translating back to the target language. (iv) "XLT (Eng-FT)", where XLT approach is applied to the Eng-FT model. (v)

---

[1] We adopt statistics on CC-MAIN-2023-23: https://commoncrawl.github.io/cc-crawl-statistics/

Table 1: Zero-shot prompts and number of testing instances for each dataset.

| Task | # Test | Zero-Shot Prompt |
|------|--------|------------------|
| MGSM | 250 | Let's think step by step. Question: {question} |
| XCOPA | 500 | Here is a premise and a question. Help me pick the more plausible option. Premise: {premise} Question: What is the {question}? (A) {choice1} (B) {choice2} |
| XNLI | 1000 | You should judge whether the hypothesis is true (entailment), false (contradiction), or undetermined (neutral) given the premise. Premise: {premise} Hypothesis: {hypothesis} |
| XQuAD | 1000 | {context} Question: {question} |
| XLSum | 500 | Summarize the context in one sentence. Title: {title} Context: {article} |

Arithmetic Merging ("AriMerge") (Zhang et al., 2023a), which is the state-of-the-art adapter merging method by arithmetic addition. (vi) MAD-X (Pfeiffer et al., 2020) decomposes language and task via independent in invertible adapters. (vii) LF-SFT (Ansell et al., 2022) adopts sparse fine-tuning on language and task respectively and directly merging via addition. [2]

**Evaluation Metrics**    For reasoning and NLU tasks, we use accuracy scores as our evaluation metric. For the summarization task, we evaluate the performance by ROUGE-L score (Lin, 2004).

**Experiment Details**    The backbone model that we use to test `AdaMergeX` is Llama2-7b (Touvron et al., 2023). To fine-tune Llama2 using LoRA and IA$^3$, we configure the target modules to include all available layers. We follow the notation of Vaswani et al. (2017). In particular, we utilize the attention layer's $\{W^Q, W^K, W^V, W^O\}$ and the feed-forward layer's $\{W_1, W_2\}$ for LoRA. For IA$^3$, we focus on $W^K$ and $W^V$ in the attention layer, as well as $W_2$ in the feed-forward layer. For the merging target modules, inspired by Geva et al. (2021) who attributes task ability to the feedword layer, we merge $\{W^Q, W^V\}$ for LoRA as we focus on language ability instead. We employ conventional causal language modeling as the reference task, where the prediction of the subsequent token is based on preceding inputs. Specifically, we generate the training set from the corpora provided by Foundation by dividing them into segments with a length of 512. There is only one hyperparameter in our method, which is $t$ in Equation ( 6) and Equation ( 8). When tuning this hyperparameter, for each task, we select the validation set from French and then extend it to encompass all other languages, for those tasks that do not contain French validation set, we adopt Vietnamese instead. For XLT method (Huang et al., 2023b), we adopt the same zero-shot prompts as in the original paper.

## 4.2    MAIN RESULTS

Table 2 presents our main experimental results on 5 representative cross-lingual tasks, where we report the average scores across all languages. Detailed results of each language are shown in Table 8 and 9 in Appendix A.3 for LoRA and (IA)$^3$ respectively.

**AdaMergeX outperforms direct transfer and prompting methods**    When comparing to fine-tuning on the task in English and direct transfer to the target language, `AdaMergX` outperforms it on all settings and achieves $1.4\%$ absolute improvement with LoRA and $1.5\%$ absolute improvement with (IA)$^3$. When comparing to the state-of-the-art method for cross-lingual transfer in LLMs via prompting, XLT with Vanilla Llama2 model ("XLT (Vanilla)") and model fine-tuned on target task in English ("XLT (Eng-FT)"), `AdaMerge` outperforms it on all settings and achieves $3.4\%$ absolute improvement with LoRA and $7.3\%$ absolute improvement with (IA)$^3$. This achievement proves that the introduction of adapter merging to achieve cross-lingual transfer is effective, especially in the circumstance of LLMs.

**AdaMergeX outperforms general adapter merging methods**    Compared with the state-of-the-art method for adapter merging namely Arimerge, `AdaMergeX` outperforms it on all settings and achieves $6.9\%$ absolute improvement with LoRA and $2.3\%$ absolute improvement with (IA)$^3$. Therefore, our adaptive merging method `AdaMerge` which considers the structure of adapters

---

[2]As MAD-X and LT-SFT are not applicable to Llama, we compare under XLM-R, as shown in Table 5.

Table 2: Main experimental results on 5 representative cross-lingual tasks. Details of the selected zero-shot prompt, the baselines, and hyperparameters are described in Section 4.1.

| Adapters | Method | Reasoning | | NLU | | NLG | Avg. |
| | | MGSM | XCOPA | XNLI | XQuAD | XLSum | |
|---|---|---|---|---|---|---|---|
| LoRA | Vanilla | 2.7 | 52.3 | 14.8 | 0.0 | 20.9 | 18.1 |
| | Eng-FT | 17.4 | 58.1 | 30.3 | 31.0 | 22.9 | 31.9 |
| | XLT(Vanilla) | 2.8 | 52.6 | 23.7 | 19.3 | 1.3 | 19.9 |
| | XLT(Eng-FT) | 18.1 | 58.2 | 27.7 | 26.4 | 19.1 | 29.9 |
| | AriMerge | 6.0 | 57.9 | 13.6 | 30.1 | 19.5 | 25.4 |
| | AdaMergeX | **19.2** | **59.0** | **33.6** | **31.6** | **23.3** | **33.3** |
| $(IA)^3$ | Vanilla | 2.7 | 52.3 | 14.8 | 0.0 | **20.9** | 18.1 |
| | Eng-FT | 2.3 | 52.5 | 26.5 | 34.0 | 17.4 | 26.5 |
| | XLT(Vanilla) | 2.8 | 52.6 | 23.7 | 19.3 | 1.3 | 19.9 |
| | XLT(Eng-FT) | 2.8 | 52.6 | 25.5 | 21.3 | 1.4 | 20.7 |
| | AriMerge | 0.7 | 51.5 | 28.2 | 32.4 | 15.5 | 25.7 |
| | AdaMergeX | **3.9** | **53.1** | **28.6** | **35.5** | 18.8 | **28.0** |

outperforms all previous methods that only adopt arithmetic addition for all kinds of adapters. In the case of $(IA)^3$, the adaptive merging method, depicted in Equation (8), is specifically designed to preserve the fine-tuned coefficients distributions. However, when it comes to LoRA, where `AdaMergeX` also utilizes element-wise addition as illustrated in Equation (6), AriMerge fails due to its limitation in arithmetic weighting, which necessitates the sum to be equal to one. This constraint undermines the fundamental assumption that language ability is not limited to a specific task. More details about the further analysis of the adaptive merging step are illustrated in Section 4.3.

**`AdaMergeX` performances consistently well with LoRA and $(IA)^3$** LoRA achieves higher absolute performance than $(IA)^3$, which shows the effectiveness of LoRA on fine-tuning. However, compared to the absolute improvement of `AdaMergeX` on LoRA and $(IA)^3$, they are comparable. For example, for MGSM, LoRA and $(IA)^3$ get the same absolute improvement $1.1\%$, and for XNLI, on which LoRA and $(IA)^3$ both achieve the highest absolute improvement, their performance are comparable. This proves that `AdaMergeX` performs consistently well on different adapters.

### 4.3 DETAILED ANALYSIS

In this section, we validate its generalizability across various aspects such as the source language, reference task, backbone model, and target modules, among others. Additionally, we conduct an analysis on `AdaMergeX` to ascertain the indispensability of the adaptive merging method.

**Source Language** To prove the generalizability of `AdaMergeX` on the source language, we explore its performance with different source languages in Table 3. We test on five source languages including German, French, Spanish, Thai, and Vietnamese. We find that the performance is highly related to the source language, which depends on the language ability of the corresponding language. However, the improvements are consistent across languages. For example, the improvement was most significant with Vietnamese as the source language, with an absolute improvement of $3.4\%$ with LoRA and $3.8\%$ with $(IA)^3$. Therefore, `AdaMergeX` consistently performs well with different source languages.

**Reference Task** To prove the generalizability of `AdaMergeX` on the reference task, we explore its performance with different reference task in Table 4. We test on three different reference tasks, including XCOPA, XNLI, XQuAD, while the source language is English. The dataset was tested on the corresponding available languages among German, French, Spanish, Thai, and Vietnamese. Specifically, the improvement was most significant with XQuAD as the reference task, with an absolute improvement of $1.3\%$ with LoRA and $1.7\%$ with $(IA)^3$. Thus, it verifies that `AdaMergeX` is general to any reference task.

**Backbone Models** Not limited to Decode-only Models such as LLama2, we do further analysis on Encoder-Decoder model T5-base (Raffel et al., 2020) to prove its universal effectiveness. `AdaMerge` achieves consistently the best performance compared to fine-tuning on English and AriMerge as

Table 3: Ablation study on source language. Each dataset is tested on the corresponding available languages among German, French, Spanish, Thai, and Vietnamese.

| Adapters | Source Language | Method | MGSM | XCOPA | XNLI | XQuAD | Avg. |
|---|---|---|---|---|---|---|---|
| LoRA | German | De-Tune | 20.9 | – | 48.3 | 44.4 | 37.9 |
| | | AdaMergeX | 22.3 | – | 50.9 | 46.5 | **39.9** |
| | French | Fr-Tune | 19.9 | – | 52.9 | – | 36.4 |
| | | AdaMergeX | 22.2 | – | 57.1 | – | **39.6** |
| | Spanish | Es-Tune | 19.2 | – | 33.9 | 45.4 | 32.8 |
| | | AdaMergeX | 18.7 | – | 35.1 | 49.1 | **34.3** |
| | Thai | Th-Tune | 3.2 | 49.3 | 1.9 | 39.8 | 23.6 |
| | | AdaMergeX | 4.5 | 48.9 | 6.2 | 44.2 | **26.0** |
| | Vietnamese | Vi-Tune | – | 63.8 | 49.1 | 36.2 | 49.7 |
| | | AdaMergeX | – | 64.2 | 53.2 | 38.9 | **52.1** |
| $(IA)^3$ | German | De-Tune | 2.9 | – | 43.5 | 45.6 | 30.7 |
| | | AdaMergeX | 6.3 | – | 44.0 | 47.1 | **32.5** |
| | French | Fr-Tune | 2.5 | – | 48.7 | – | 25.6 |
| | | AdaMergeX | 4.1 | – | 47.9 | – | **26.0** |
| | Spanish | Es-Tune | 3.5 | – | 49.2 | 45.9 | 32.9 |
| | | AdaMergeX | 5.3 | – | 50.9 | 44.6 | **33.6** |
| | Thai | Th-Tune | 1.2 | 49.8 | 0.0 | 27.7 | 19.7 |
| | | AdaMergeX | 1.9 | 50.4 | 0.0 | 28.9 | **20.3** |
| | Vietnamese | Vi-Tune | – | 49.8 | 45.5 | 33.2 | 42.8 |
| | | AdaMergeX | – | 48.7 | 50.2 | 36.1 | **45.0** |

Table 4: Ablation study on reference Task. Each dataset is tested on the corresponding available languages among German, French, Spanish, Thai, and Vietnamese.

| Adapters | Reference Task | Method | MGSM | XCOPA | XNLI | XQuAD | Avg. |
|---|---|---|---|---|---|---|---|
| LoRA | – | Eng-Tune | 14.4 | 59.9 | 44.6 | 42.3 | 40.3 |
| | XCOPA | AdaMergeX | 15.2 | 60.2 | 45.1 | 43.8 | 41.1 |
| | XNLI | AdaMergeX | 14.5 | 60.9 | 46.7 | 44.1 | **41.6** |
| | XQuAD | AdaMergeX | 14.9 | 61.8 | 45.4 | 44.4 | **41.6** |
| $(IA)^3$ | – | Eng-Tune | 2.6 | 52.7 | 40.0 | 39.2 | 33.6 |
| | XCOPA | AdaMergeX | 4.9 | 54.3 | 40.5 | 40.4 | 35.0 |
| | XNLI | AdaMergeX | 3.6 | 54.6 | 41.2 | 39.9 | 34.8 |
| | XQuAD | AdaMergeX | 4.1 | 53.9 | 42.1 | 41.0 | **35.3** |

shown in Table 10 of Appendix A.4. In addition, we also implement our method on Encoder-only model XLM-R and compare with MAD-X and LF-SFT as shown in Table 5. This shows the flexibility of choosing the backbone model when implementing AdaMergeX. Moreover, it demonstrates a significant improvement when compared to MAD-X and LF-SFT, thus validating our decomposition of task ability and language ability.

Table 5: Ablation study on backbone models with XLM-Roberta-base.

| Task | Method | tr | vi | th | sw | el | ru | Avg. |
|---|---|---|---|---|---|---|---|---|
| XCOPA | MAD-X | 60.3 | 66.1 | 61.8 | 56.3 | - | - | 59.5 |
| | AdaMergeX | 69.4 | 70.5 | 66.9 | 63.2 | - | - | **67.5** |
| XNLI | MAD-X | - | - | 54.3 | 57.8 | 55.7 | 51.1 | 54.7 |
| | LF-SFT | - | - | 65.5 | 64.6 | 75.2 | 58.6 | 66.0 |
| | AdaMergeX | - | - | 70.2 | 70.4 | 77.9 | 63.8 | **70.6** |

**Merging Method** We conduct an ablation analysis on merging method to ascertain the indispensability and the effectiveness of adaptive merging in `AdaMergeX`. Table 11 in Appendix A.5 shows the detailed results, where `AdaMergeX` (adaptive) represents `AdaMergeX` with adaptive merging methods, while `AdaMergeX` (cross) represents `AdaMergeX` with cross merging methods, i.e., LoRA with merging method of $(IA)^3$ and vice versa. We find that when applying the merging method of $(IA)^3$ to LoRA, the performance is reduced much, and vice versa. Detailed analysis is shown in Appendix A.5. As a result, the adaptive merging method is crucial for adapter merging.

**Hyperparameter** $t$ Hyperparameter $t$ in Equation (6) and (8) are significant to `AdaMergeX`. Figure 2 and Figure 3 in Appendix A.6 show the performance of `AdaMergeX` on XNLI, MGSM and XLSum as the change of $t$. We find that $t$ has less influence on $(IA)^3$ compared to LoRA. Furthermore, in the circumstance of LoRA, the trend of the influence depends highly on the task. XNLI benefits from higher $t$ value, while MGSM benefits from lower $t$ value.

## 5 RELATED WORK

### 5.1 CROSS-LINGUAL TRANSFER

The emergence of multilingual systems (Kenton & Toutanova, 2019; Conneau & Lample, 2019; Conneau et al., 2020; OpenAI, 2022; Anil et al., 2023; Touvron et al., 2023) has sparked interest in cross-lingual transfer (Kim et al., 2017; Lin et al., 2019; Schuster et al., 2019; Pfeiffer et al., 2020). Fine-tuning on the target language and target task is an intuitive way to make models obtain the ability of this task, but it is too costly in the era of LLMs as we always lack enough training data (Ma et al., 2023). Alternatively, some researchers explore realigning representations among languages (Nguyen et al., 2023; Salesky et al., 2023; Gao et al., 2023). However, Gaschi et al. (2023) demonstrates that aligned representations do not significantly benefit cross-lingual transfer. To address this issue, some works adopt explicit translation to achieve cross-lingual transfer (Liang et al., 2023; Huang et al., 2023b). However, they rely on translation ability which is not guaranteed. Furthermore, in the era of in-context learning (Brown et al., 2020; Chowdhery et al., 2022; Touvron et al., 2023; OpenAI, 2023), Li et al. (2023) and Tanwar et al. (2023) utilize prompt tuning to achieve cross-lingual transfer. Nevertheless, the performance remains limited for low-resource languages, which is often not carefully considered in the pre-training of LLMs.

### 5.2 MODEL MERGING

Model merging has been widely used in image identification (Wortsman et al., 2022; Matena & Raffel, 2022), knowledge editing (Mitchell et al., 2022; Meng et al., 2022) and task combination (Ilharco et al., 2022). In the era of PEFT, researchers have started exploring different approaches to merging adapters (Zhang et al., 2023a; Yadav et al., 2023; Huang et al., 2023a; Chronopoulou et al., 2023; Ponti et al., 2023). These studies, however, have primarily focused on task transfer and have solely utilized linear combinations, which may not be applicable to all types of adapters. Therefore, there is a need for further investigation and development of more general merging techniques that can be applied to a wider range of adapter types.

## 6 CONCLUSION

In this work, we propose a new cross-lingual transfer method `AdaMergeX`. We split target task ability in the target language into two parts: "task ability" and "language ability". In the context of PEFT, task ability can be obtained by tuning on the target task in the source language. To achieve cross-lingual transfer, which aims to transfer task ability from the source language to the target language, we introduce a reference task from which we obtain language ability and further merge it to task ability by adapter merging. Different from all previous adapter merging methods, we propose a structure adaptive adapter merging method that aligns the adapter merging method with the way adapters combined to LLMs. Experiment results show that `AdaMergeX` performs well among all settings. Moreover, ablation analysis proves that `AdaMergeX` is robust to backbone models, source languages, and source tasks, which further demonstrates its usability.

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

# A  APPENDIX

## A.1  ADAMERGEX ON PREFIC-TUNING

The results demonstrate that `AdaMergeX` excels remarkably within the realm of prefix-tuning, a distinct and separate approach to fine-tuning. Results on XNLI task with mT5 (Xue et al., 2021) are shown as follows.

| Method | es | fr |
|---|---|---|
| Eng-FT | 31.2 | 29.7 |
| AriMerge | 29.8 | 28.3 |
| AdaMergeX | 34.1 | 31.4 |

Table 6: Results of `AdaMergeX` on Prefix-tuning

## A.2  PROMPTS

---

**MGSM (French)**

---

Let's think step by step.

Question: Les canes de Janet pondent 16 œufs par jour. Chaque matin, elle en mange trois au petit déjeuner et en utilise quatre autres pour préparer des muffins pour ses amis. Ce qui reste, elle le vend quotidiennement au marché fermier, au prix de 2 $ l'œuf de cane frais. Combien (en dollars) gagne-t-elle chaque jour au marché fermier ?
Answer:

---

**XCOPA (Vietnamese)**

---

Here is a premise and a question. Help me pick the more plausible option. Answer with (A) or (B).

Premise: Các mt hàng đã đc đóng gói trong bc bong bóng.
Question: What is the cause?
(A) Nó d v.
(B) Nó nh.
Answer:

---

**XNLI (French)**

---

You should judge whether the hypothesis is true (entailment), false (contradiction), or undetermined (neutral) given the premise. The relationship can be chosen from entailment, contradiction, and neutral.

Premise: Cela fait 17 ans que je suis affilié à l'IRT.
Hypothesis: Je n'ai rien à voir avec l'IRT.
Relationship:

---

**XLSum (Vietnamese)**

---

Summarize the context in one sentence.

Title: Côte d'Ivoire : le groupe Magic System fête ses 20 ans
Context: Formé en 1997, le groupe a connu la consécration deux ans plus tard avec son tube Premier Gaou. Le groupe ivoirien fête ses 20 ans avec une tournée africaine et une autobiographie. Ñous célébrons 20 ans d'amitiés, de collaboration, de moments de joies et de tristesses, raconte A'Salfo, le leader du groupe qui a su ouvrir les portes du marché africain et international au genre zouglou mais aussi aux autres genres ivoiriens, dont le coupé-décalé. A'Salfo, Manadja, Tino et Goudé, les quatre boys d'Anoumabo, quartier déshérités d'Abidjan, aux ruelles boueuses et sablonneuses, ont joué partout, des stades africains aux salles mythiques comme l'Apollo à New York ou l'Olympia à Paris et jusqu'au Louvre, le 7 mai, pour le concert célébrant la victoire du président français Emmanuel Macron. Magic System a bénéficié de conseils avisés d'Alpha Blondy. Formé en 1997, le groupe a connu la consécration deux ans plus tard avec son tube Premier Gaou, fable sur les déboires sentimentaux d'un jeune homme naïf - le gaou est un homme crédule en nouchi, l'argot abidjanais. Le tube va propulser les quatre amis sur la scène mondiale. Magic System a multiplié les succès, enchaînant les albums, sans oublier l'amitié. Ñagic System est aussi un groupe qui a toujours voulu relever les défis, après Premier Gaou, nos détracteurs ont parlé de coup de chance! On a donc relevé ce défi, explique Manadja, le ğrosğu groupe. Le groupe reconnaît avoir bénéficié de conseils avisés, dont ceux de la star ivoirienne du reggae, Alpha Blondy.
Summary:

---

**XQuAD (French)**

---

Ni mà din tích mt ct ngang liên quan đn khi lng mà ten-x ng sut đc tính toán. Hình thc này bao gm thut ng áp sut gn lin vi các lc hot đng bình thng đi vi khu vc ct ngang (đng chéo ma trn ca tenx) cũng nh các thut ng ct gn lin vi các lc tác đng song song vi din tích mt ct ngang (các yu t ngòai đng chéo). Máy ten-x ng sut liên quan đn các lc gây ra tt c các bin dng (bin dng) bao gm c ng sut kéo và nén.:133–134:38-1–38-11

Question: Điu gì đc s dng đ tính din tích mt ct trong th tích ca mt vt th?
Answer:

---

Table 7: One-shot prompting examples of tested datasets.

| Models | Method | de | ru | fr | es | zh | vi | tr | ar | el | th | hi | sw |
|--------|--------|-----|-----|-----|-----|-----|-----|-----|-----|-----|-----|-----|-----|
| MGSM | Vanilla | 2.4 | 3.6 | 3.6 | 3.2 | 2.4 | – | – | – | – | 2.0 | – | 2.0 |
| | Eng-FT | 22.4 | 24.8 | 20.4 | 22.4 | 22.8 | – | – | – | – | 6.8 | – | 2.4 |
| | XLT(Vanilla) | 2.0 | 2.8 | 2.8 | 3.2 | 2.8 | – | – | – | – | 2.0 | – | 3.2 |
| | XLT(Eng-FT) | 22.0 | 24.0 | 22.8 | 24.4 | 24.2 | – | – | – | – | 5.2 | – | 4.4 |
| | AriMerge | 6.4 | 8.0 | 2.4 | 10.4 | 3.2 | – | – | – | – | 11.6 | – | 0.0 |
| | AdaMergeX | 24.8 | 26.2 | 23.6 | 22.4 | 22.0 | – | – | – | – | 8.0 | – | 7.2 |
| XCOPA | Vanilla | – | – | – | – | 54.4 | 54.0 | – | – | – | 51.8 | – | 49.0 |
| | Eng-FT | – | – | – | – | 61.8 | 67.2 | – | – | – | 52.6 | – | 50.6 |
| | XLT(Vanilla) | – | – | – | – | 56.8 | 52.4 | – | – | – | 51.0 | – | 50.0 |
| | XLT(Eng-FT) | – | – | – | – | 60.6 | 70.0 | – | – | – | 51.6 | – | 50.4 |
| | AriMerge | – | – | – | – | 61.0 | 69.8 | – | – | – | 50.6 | – | 50.0 |
| | AdaMergeX | – | – | – | – | 61.8 | 69.8 | – | – | – | 51.8 | – | 52.2 |
| XNLI | Vanilla | 27.4 | 26.6 | 24.0 | 20.2 | 0.3 | 21.5 | 14.3 | 0.1 | 0.3 | 0.3 | 0.0 | 43.0 |
| | Eng-FT | 54.0 | 54.0 | 58.2 | 60.5 | 33.5 | 47.0 | 9.6 | 0.8 | 5.4 | 3.3 | 5.2 | 31.8 |
| | XLT(Vanilla) | 44.7 | 44.4 | 39 | 36.9 | 5.3 | 36 | 20.6 | 0.4 | 0.2 | 13.9 | 0.2 | 42.6 |
| | XLT(Eng-FT) | 54.1 | 44.3 | 44.6 | 58.6 | 34.0 | 43.0 | 15.9 | 0.0 | 1.2 | 2.0 | 0.9 | 33.9 |
| | AriMerge | 28.7 | 16.5 | 12.8 | 21.2 | 1.0 | 32.1 | 16.2 | 0.3 | 1.8 | 0.0 | 10.2 | 22.8 |
| | AdaMergeX | 57.8 | 56.7 | 63.1 | 62.8 | 32.9 | 49.2 | 10.3 | 1.0 | 9.1 | 13.3 | 14.9 | 35.9 |
| XLSum | Vanilla | – | 13.4 | 12.5 | 11.4 | 56.0 | 22.1 | 15.7 | 23.5 | – | 14.8 | 31.6 | 8.1 |
| | Eng-FT | – | 21.7 | 16.1 | 11.3 | 58.4 | 21.2 | 16.4 | 25.8 | – | 15.6 | 32.9 | 9.9 |
| | XLT(Vanilla) | – | 0.6 | 2.3 | 1.8 | 0.5 | 1.3 | 0.8 | 0.8 | – | 0.2 | 0.8 | 2.1 |
| | XLT(Eng-FT) | – | 17.8 | 5.0 | 6.6 | 56.8 | 13.5 | 10.8 | 28.9 | – | 13.5 | 33.9 | 3.9 |
| | AriMerge | – | 14.5 | 8.7 | 9.8 | 49.8 | 12.6 | 11.7 | 29.8 | – | 17.2 | 34.2 | 6.5 |
| | AdaMergeX | – | 21.6 | 16.2 | 11.9 | 58.4 | 21.6 | 16.7 | 25.6 | – | 15.5 | 33.9 | 11.4 |
| XQuAD | Vanilla | 0.0 | 0.0 | – | 0.0 | 0.0 | 0.0 | 0.0 | 0.0 | 0.0 | 0.0 | 0.0 | – |
| | Eng-FT | 49.0 | 34.1 | – | 48.2 | 53.5 | 40.9 | 17.3 | 10.2 | 13.9 | 31.0 | 11.8 | – |
| | XLT(Vanilla) | 34.8 | 14.0 | – | 29.8 | 33.1 | 21.8 | 20.2 | 12.0 | 8.6 | 7.1 | 12.1 | – |
| | XLT(Eng-FT) | 39.1 | 26.3 | – | 40.7 | 41.2 | 33.9 | 19.0 | 13.8 | 13.0 | 23.8 | 13.2 | – |
| | AriMerge | 50.7 | 31.8 | – | 49.1 | 50.2 | 42.3 | 15.9 | 10.4 | 12.6 | 28.7 | 9.7 | – |
| | AdaMergeX | 50.7 | 34.1 | – | 50.0 | 53.2 | 41.7 | 17.3 | 10.4 | 13.7 | 31.8 | 13.1 | – |

Table 8: Comprehensive experimental results for both baselines and AdaMergeX are obtained across all datasets in corresponding available languages. The fine-tuning method employed was LoRA, with Llama2-7b serving as the backbone model.

## A.3 DETAILED RESULTS

## A.4 ADAMERGEX ON T5-BASE

In the case of LoRA on XNLI, AdaMergeX obtains $4.2\%$ absolute improvements in Spanish and $2.8\%$ absolute improvements in French. For $(IA)^3$, the improvements are $1.1\%$ and $4.0\%$ respectively.

## A.5 ABLATION ON ADAPTIVE MERGING

We find that when applying the merging method of $(IA)^3$ to LoRA, the performance is reduced much. Specifically, on XNLI the performance gets $39.5\%$ absolute reduction, while for XQuAD the reduction is $45.9\%$ absolute value. When applying the merging method of LoRA to $(IA)^3$, the performance also decreases compared to that of the adaptive merging method. For XNLI the reduction is $2.4\%$, while for XQuAD the reduction is $0.7\%$. The reduction is smaller than that for LoRA. This can be attributed to the fact that the fine-tuning of $(IA)^3$ is not as effective as that of LoRA and has a relatively minor impact on the overall model performance.

## A.6 ABLATION ON HYPER-PARAMETER $t$

## A.7 ABLATION ON MERGING MODULES

| Models | Method | de | ru | fr | es | zh | vi | tr | ar | el | th | hi | sw |
|--------|--------|----|----|----|----|----|----|----|----|----|----|----|----|
| MGSM | Vanilla | 2.4 | 3.6 | 3.6 | 3.2 | 2.4 | – | – | – | – | 2.0 | – | 2.0 |
| | Eng-FT | 2.0 | 2.0 | 3.6 | 2.4 | 1.6 | – | – | – | – | 2.4 | – | 2.0 |
| | XLT(Vanilla) | 2.0 | 2.8 | 2.8 | 3.2 | 2.8 | – | – | – | – | 2.0 | – | 3.2 |
| | XLT(Eng-FT) | 0.8 | 1.6 | 4.8 | 4.0 | 3.2 | – | – | – | – | 2.8 | – | 2.4 |
| | AriMerge | 0.0 | 0.4 | 0.4 | 0.0 | 1.6 | – | – | – | – | 2.0 | – | 0.4 |
| | AdaMergeX | 4.4 | 3.6 | 4.8 | 6.0 | 3.6 | – | – | – | – | 2.8 | – | 2.0 |
| XCOPA | Vanilla | – | – | – | – | 54.4 | 54.0 | – | – | – | 51.8 | – | 49.0 |
| | Eng-FT | – | – | – | – | 54.8 | 54.2 | – | – | – | 51.2 | – | 49.8 |
| | XLT(Vanilla) | – | – | – | – | 56.8 | 52.4 | – | – | – | 51.0 | – | 50.0 |
| | XLT(Eng-FT) | – | – | – | – | 56.8 | 53.2 | – | – | – | 51.4 | – | 49.8 |
| | AriMerge | – | – | – | – | 53.0 | 50.6 | – | – | – | 52.2 | – | 50.2 |
| | AdaMergeX | – | – | – | – | 55.0 | 55.2 | – | – | – | 52.1 | – | 50.0 |
| XNLI | Vanilla | 27.4 | 26.6 | 24.0 | 20.2 | 0.3 | 21.5 | 14.3 | 0.1 | 0.3 | 0.3 | 0.0 | 43.0 |
| | Eng-FT | 46.4 | 45.3 | 51.9 | 50.7 | 1.6 | 51.0 | 31.4 | 0.1 | 0.8 | 0.0 | 0.0 | 39.3 |
| | XLT(Vanilla) | 44.7 | 44.4 | 39.0 | 36.9 | 5.3 | 36.0 | 20.6 | 0.4 | 0.2 | 13.9 | 0.2 | 42.6 |
| | XLT(Eng-FT) | 34.3 | 36.8 | 36.3 | 34.2 | 25.4 | 34.4 | 32.1 | 5.2 | 3.8 | 20.7 | 8.0 | 34.4 |
| | AriMerge | 42.4 | 47.2 | 52.9 | 49.3 | 6.4 | 54.5 | 49.1 | 0.2 | 0.5 | 0.1 | 0.0 | 35.5 |
| | AdaMergeX | 45.3 | 46.5 | 53.0 | 54.3 | 1.5 | 58.8 | 41.7 | 2.2 | 0.9 | 0.1 | 0.1 | 38.4 |
| XLSum | Vanilla | – | 13.4 | 12.5 | 11.4 | 56.0 | 22.1 | 15.7 | 23.5 | – | 14.8 | 31.6 | 8.1 |
| | Eng-FT | – | 4.2 | 9.0 | 6.8 | 56.6 | 14.7 | 13.6 | 16.6 | – | 12.5 | 32.3 | 7.6 |
| | XLT(Vanilla) | – | 0.6 | 2.3 | 1.8 | 0.5 | 1.3 | 2.5 | 0.8 | – | 0.2 | 0.8 | 2.1 |
| | XLT(Eng-FT) | – | 0.6 | 3.1 | 1.8 | 0.4 | 1.3 | 2.5 | 1.1 | – | 0.3 | 0.8 | 2.1 |
| | AriMerge | – | 4.8 | 6.3 | 7.6 | 44.1 | 9.9 | 11.8 | 15.4 | – | 13.1 | 32.3 | 9.4 |
| | AdaMergeX | – | 6.8 | 10.5 | 7.5 | 55.2 | 14.9 | 15.3 | 23.5 | – | 13.6 | 33.4 | 7.7 |
| XQuAD | Vanilla | 0.0 | 0.0 | – | 0.0 | 0.0 | 0.0 | 0.0 | 0.0 | 0.0 | 0.0 | 0.0 | – |
| | Eng-FT | 47.3 | 32.8 | – | 47.6 | 53.7 | 35.1 | 28.9 | 22.8 | 21.9 | 26.9 | 23.2 | – |
| | XLT(Vanilla) | 34.8 | 14.0 | – | 29.8 | 33.1 | 21.8 | 20.2 | 12.0 | 8.6 | 7.1 | 12.1 | – |
| | XLT(Eng-FT) | 37.1 | 16.8 | – | 32.4 | 37.6 | 25.1 | 19.3 | 14.0 | 10.0 | 7.0 | 14.1 | – |
| | AriMerge | 46.0 | 32.2 | – | 44.5 | 51.2 | 35.4 | 28.2 | 23.4 | 20.6 | 21.6 | 20.7 | – |
| | AdaMergeX | 48.6 | 33.0 | – | 48.2 | 56.0 | 35.7 | 29.3 | 25.4 | 24.5 | 29.2 | 24.6 | – |

Table 9: Comprehensive experimental results for both baselines and AdaMergeX are obtained across all datasets in corresponding available languages. The fine-tuning method employed was (IA)[3], with Llama2-7b serving as the backbone model.

Table 10: Ablation study on backbone models. Results are evaluated on T5-base.

| Adapters | Method | XNLI | |
|----------|--------|------|------|
| | | es | fr |
| LoRA | Eng-FT | 33.0 | 32.9 |
| | AriMerge | 34.1 | 30.1 |
| | AdaMergeX | **37.2** | **35.7** |
| (IA)[3] | Eng-FT | 38.2 | 38.4 |
| | AriMerge | 35.6 | 36.1 |
| | AdaMergeX | **39.3** | **42.4** |

Table 11: Ablation study on adaptive merging method. `AdaMergeX (adaptive)` represents `AdaMergeX` with adaptive merging methods, while `AdaMergeX (cross)` represents `AdaMergeX` with cross merging methods, i.e., LoRA with merging method of (IA)$^3$ and vice versa. Increase ↑ and decrease ↓ are both compared to the baseline method Eng-Tune.

| Adapters | Tasks | Method | es | vi | Avg. |
|---|---|---|---|---|---|
| LoRA | XNLI | Eng-Tune | 60.5 | 47.0 | 53.8 |
| | | AdaMergeX (adaptive) | **62.8** ↑ 2.3 | **49.2** ↑ 2.2 | **56.0** ↑ 2.2 |
| | | AdaMergeX (cross) | 17.6 ↓ 42.9 | 15.4 ↓ 31.6 | 16.5 ↓ 37.3 |
| | XQUAD | Eng-Tune | 48.2 | 40.9 | 44.6 |
| | | AdaMergeX (adaptive) | **50.0** ↑ 1.8 | **41.7** ↑ 0.8 | **45.9** ↑ 1.3 |
| | | AdaMergeX (cross) | 0.0 ↓ 48.2 | 0.0 ↓ 40.9 | 0.0 ↓ 44.6 |
| (IA)$^3$ | XNLI | Eng-Tune | 50.7 | 51.0 | 50.9 |
| | | AdaMergeX (adaptive) | **54.3** ↑ 3.6 | **58.8** ↑ 7.8 | **56.4** ↑ 5.5 |
| | | AdaMergeX (cross) | 50.9 ↑ 0.2 | 57.4 ↑ 6.4 | 54.2 ↑ 3.1 |
| | XQUAD | Eng-Tune | 47.6 | 35.1 | 41.4 |
| | | AdaMergeX (adaptive) | **48.2** ↑ 0.6 | **35.7** ↑ 0.6 | **42.0** ↑ 0.6 |
| | | AdaMergeX (cross) | 47.5 ↓ 0.1 | 34.9 ↓ 0.2 | 41.3 ↓ 0.1 |

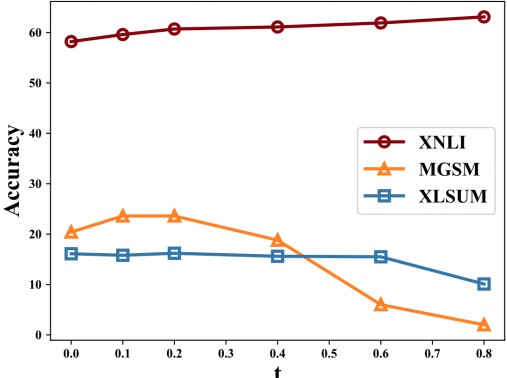 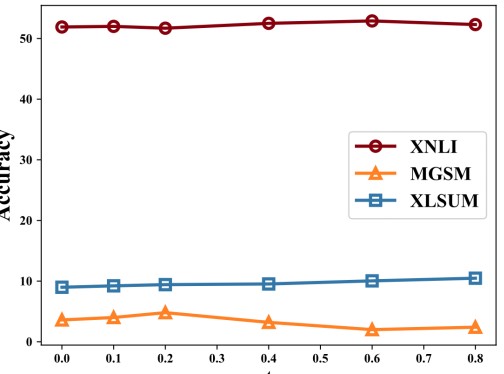

Figure 2: Impact of $t$ in LoRA for different tasks.  Figure 3: Impact of $t$ in (IA)$^3$ for different tasks.

| Models | Method | de | ru | fr | es | th | sw | Avg. |
|---|---|---|---|---|---|---|---|---|
| XNLI | Eng-Tune | 63.3 | 56.4 | 56.6 | 58.6 | 4.1 | 41.5 | 46.8 |
| | AdaMergeX | 63.8 | 57.2 | 58.2 | 58.9 | 3.7 | 41.8 | **47.3** ↑ 0.5 |
| XQuAD | Eng-Tune | 9.8 | 8.7 | — | 15.2 | 4.4 | — | 9.5 |
| | AdaMergeX | 10.4 | 7.8 | — | 21.4 | 5.4 | — | **11.2** ↑ 1.7 |

Table 12: Llama2-7b on LoRA with fine-tuning target modules as $W^Q, W^V$ and merging target modules as $W^Q, W^V$.

| Models | Method | de | ru | fr | es | th | sw | Avg. |
|---|---|---|---|---|---|---|---|---|
| XNLI | Eng-Tune | 54.0 | 54.0 | 58.2 | 60.5 | 3.3 | 31.8 | 43.6 |
| | AdaMergeX | 53.7 | 55.6 | 60.5 | 62.7 | 4.9 | 33.6 | **45.2** ↑ 1.6 |
| XQuAD | Eng-Tune | 49.0 | 34.1 | — | 48.2 | 31.0 | — | 40.6 |
| | AdaMergeX | 50.2 | 32.9 | — | 48.9 | 31.3 | — | **40.8** ↑ 0.2 |

Table 13: Llama2-7b on LoRA with fine-tuning target modules as $W^Q, W^K, W^V, W^O, W_1, W_2$ and merging target modules as $W^Q, W^K, W^V, W^O, W_1, W_2$.

