# OpenReview forum: "Cross-Lingual Transfer with Large Language Models via Adaptive Adapter Merging"
_ICLR.cc/2024/Conference — Submitted to ICLR 2024_

### Official Review · Reviewer_PiPR · 2023-10-27

**Soundness:** 2 fair
**Presentation:** 2 fair
**Contribution:** 2 fair
**Rating:** 3
**Confidence:** 4

**Summary:**

This paper studies model merging in cross-lingual transfer. The authors propose learning adapters that capture 'task ability' and 'language ability' separately and introduce a novel adaptive adapter merging method called AdaMergeX, which is applied to LoRA adapters and IA3 adapters. The core idea of the paper is to achieve cross-lingual *task* transfer with merging of 3 adapters, namely 2 reference task adapters (1 in source language, 1 in target language), 1 task adapter (in source language for the target task). Element-wise addition/subtraction is applied to LoRA type adapters and element-wise multiplication is applied to IA3 adapters.
The paper examines the proposed method in several cross-lingual transfer tasks and demonstrates that it achieves improved performance.

**Strengths:**

The paper conducted experiments on a range of cross-lingual transfer tasks for LoRA and IA3. Additionally, the paper proposed an interesting distinction: merging different types of adapters may require different operations.

**Weaknesses:**

There are several drawbacks in the proposed work:
- Inadequate benchmark / baselines, for example:
    * The paper argues that the lack of enough training data to study standard adapters (Houlsby) merging is unconvincing. (See MAD-X as an example, which is not fundamentally different in terms of training data requirements compared to Houlsby nor this work).
     * The test results are sub-sampled (in paper, "while for XNLI, XLSum, and XQuAD we randomly sample 1000, 500,
and 1000 data points from the whole test set respectively") without convincing justifications.

- Ablations:
    * Even though differences in results between the same task across two languages approximate 'language ability,' such a claim was not carefully examined or discussed. Are the domains of training data the same for two languages (for the same task)? Was any of the cross-lingual data for the reference task machine-translated?
    * Why is the ablation of backbone models (Table 6) evaluated on only a single task with two languages? Given that XNLI contains 15 languages, it's unconvincing that the results generalize across languages, especially when prior tables (e.g., Table 2 or 4) show results across multiple languages.
    * The same question applies to Table 3, where results are only shown for 2 languages (es, fr), and they are not even the same languages as in Table 6 (es, vi). No justifications are provided.

- Inadequate discussions about the proposed work and its relationship to prior work. For example:
    * the training of adapters capturing language ability skills using 'LM objectives' (equivalent to the reference task in the paper) lacks a clear connection to existing literature in cross-lingual transfer, such as MAD-X and LT-SFT.
    * insufficient exploration of its relationship and differences with methods like AdaMerge and Task Arithmetics. The LoRA merging in the proposed work involves element-wise addition/subtraction, which is the same as in Task Arithmetics.
    * imho, one of the core interesting point proposed by the author is different types of adapters requires different merging operation, yet this aspect is not sufficiently studied in the paper.

- Writing:
    * Details of the experiments, such as the backbone model used for the experiments, are scattered throughout the paper, making it very difficult to follow the experiments.
    * Definition of AdaMergeX (adaptive) vs AdaMergeX (cross), Eng-Tune vs Eng-FT etc.
    * There are unclear descriptions of experimental settings, such as of what has been used as reference tasks for specific experiments, including details on the amount of data used and the training process etc.

**Questions:**

* Why do you randomly sub sample test set for evaluation for  XNLI, XLSum, and XQuAD? What's wrong with evaluating on all test data?
* What hyper-parameters do you use for training?
* What's the reason behind using Llama-2 or T5 as the backbone, where other backbones, especially multilingual backbones are available?


-------------------

Dear authors,
Thank you very much for the additional information. I acknowledged that I've reviewed the rebuttal and updated information.
Best,

---

> ### Author Response · Authors · 2023-11-20
> **Response to PiPR**
>
> We thank the reviewer for your time and thorough response. We address the weaknesses and concerns of the reviewer below.
>
>
>
> W1.1: Standard Adapters
>
> Thanks for the recommendation, yes, the training data is sufficient in the context of smaller models instead of LLMs such as Llama2. We have implemented it, please refer to Response 2 in the Novelty part in the general response.
>
>
>
> W1.2 & Q1: Sub-sampled testset
>
> We include the entire test set in our evaluation for MGSM and XCOPA. However, for XNLI, XLSum, and XQuAD, we employ a sampling approach. Specifically, we randomly select 1000, 500, and 1000 data points, respectively, from the complete test set. This sampling strategy is necessary due to the extended inference time of the Large Language Model (LLM) on these datasets. Without sampling, testing one method on the XNLI dataset alone would take approximately 120 hours. Therefore, it has become a common practice, as demonstrated in previous works [1] [2] [3] [4] to evaluate LLM methods on a subset of the test set. It is important to note that the sampled test set is sufficiently large to avoid any potential bias.
>
>
>
> W2.1: 'language ability' is not clearly explained
>
> Regarding the training data, there is no restriction for it to be strictly language-parallel. The only requirement is that the data should pertain to the same task. In the case of the language modeling task, the training data consists of Wikipedia corpora specific to each corresponding language, and the task is conventional language modeling, i.e., next token prediction.
>
>
>
> W2.2 & 2.3: Ablation on source languages and source tasks
>
> Please refer to Response 1 and Response 2 in the Experiment part in the general response.
>
>
>
> W3: Inadequate discussions about the proposed work and its relationship to prior work.
>
> Please refer to Response 1 and Response 2 in the Novelty part in the general response.
>
>
>
> W4: Writing
>
> Sorry for the confusion, we have addressed these issues in our new pdf version.
>
>
>
> Q2: Hyper-paramters for training
>
> Thanks for pointing out. We have listed hyper-paramters for training in the appendix of the new pdf version.
>
>
>
> Q3: Multi-Lingual Models
>
> Please refer to Response 3 in the Experiment part in the general response.
>
>
> [1] Pryzant et al, Automatic Prompt Optimization with "Gradient Descent" and Beam Search, Arxiv 2023.
>
> [2] Zhou et al, Large Language Models Are Human-Level Prompt Engineers, ICLR 2023.
>
> [3] Gonen et al, Demystifying Prompts in Language Models via Perplexity Estimation, EMNLP 2023.
>
> [4] We et al, Self-Adaptive In-Context Learning: An Information Compression Perspective for In-Context Example Selection and Ordering, ACL 2023.

---

> ### Author Response · Authors · 2023-11-22
> **Rquest more discussion with Reviewer PiPR**
>
> Dear Reviewer PiPR,
>
> We appreciate you taking the time to review our work and your insightful feedback. We would like to discuss this further with you to see if there are any unresolved questions. Specifically, we have implemented our method on the following points.
> (1) We applied our method to more adapters.
> (2) We conducted more ablation analysis on source language and reference tasks.
> (3) Thanks for your suggestions on the writing part, we have updated the new PDF version.
> If you have any further concerns, please let us know.
>
> Sincerely,
> Authors

---

### Official Review · Reviewer_tcUW · 2023-11-01

**Soundness:** 2 fair
**Presentation:** 3 good
**Contribution:** 2 fair
**Rating:** 3
**Confidence:** 5

**Summary:**

The paper proposes an adaptive adapter merging method, termed AdaMergeX, for cross-lingual transfer of large language models (LLMs). The authors decompose the abilities of multilingual LLMs into "task ability" and "language ability". AdaMergeX introduces three types of adapters to LLMs, and models the task ability and language ability by different adapter merging strategies of the three adapters. They conduct experiments on five multilingual tasks, and observe improvement over several baselines.

**Strengths:**

- The idea of adapter merging and ability composite is nice, and it could be useful for efficient cross-lingual transfer for LLMs, which are expensive to fine-tune.
- The structured-adaptive merging methods consider the structure of adapters, and consistently outperforms a strong adapter merging baseline, AriMerge.
- The authors conduct experiments on five multilingual datasets, covering reasoning, natural language understanding, and natural language generation tasks.

**Weaknesses:**

- The experimental setup is unclear and confusing. (1) Training data: Since the proposed methods learn task adapters, I would guess it learns on some training data of the downstream tasks. However, Table 1 only provides the details of test data, and it is unclear why the evaluation is conducted on small subsets of the test sets. (2) Baseline setup:  MAD-X[1] adopts a similar idea, and introduces task and language adapters to cross-lingual transfer, which should be the most important baseline. Besides, fine-tune-based cross-lingual transfer methods such as xTune[2], and translate-test[3] should be considered. The setup of the XLT baseline is not clear. If the proposed method uses training data, is XLT evaluated in a few-shot setup as well?
- The results are insufficient to support the claim "AdaMergeX consistently outperforms other state-of-the-art methods". First, the XNLI accuracy scores in Table 2 are too low to compete with random guess, which has 33.3% accuracy. For reference, mT5[4] achieves 85.0 accuracy on zero-shot cross-lingual transfer on XNLI. Besides, regarding XLT as a SOTA cross-lingual transfer method is misleading because a lot of cross-lingual transfer methods achieve better performance.
- "AdaMerge outperforms cross-lingual transfer methods" is misleading. I would guess the paper focuses on some efficient-training setup, but the results in Table 1 cannot support this claim.

[1] MAD-X: An Adapter-Based Framework for Multi-Task Cross-Lingual Transfer

[2] Consistency Regularization for Cross-Lingual Fine-Tuning

[3] Revisiting Machine Translation for Cross-lingual Classification

[4] mT5: A massively multilingual pre-trained text-to-text transformer

**Questions:**

- In Eq.3 and 4, how the symbol "~" is converted to "="?
- What is the relation between AdaMergeX and MAD-X?

---

> ### Author Response · Authors · 2023-11-20
> **Response to tcUW**
>
> We thank the reviewer for your time and thorough response. We address the weaknesses and concerns of the reviewer below.
>
>
>
> W1.1: Training data
>
> For training data of each dataset on the source language (mostly in English), we use default training data of each dataset.
>
> | Dataset   | MGSM | XCOPA | XNLI | XQuAD | XLSum |
> | --------- | ---- | ----- | ---- | ----- | ----- |
> | #Training | 7.5k | 0.5k  | 10k  | 1.2k  | 10k   |
>
>
>
> W1.1: Test data
>
> We include the entire test set in our evaluation for MGSM and XCOPA. However, for XNLI, XLSum, and XQuAD, we employ a sampling approach. Specifically, we randomly select 1000, 500, and 1000 data points, respectively, from the complete test set. This sampling strategy is necessary due to the extended inference time of the Large Language Model (LLM) on these datasets. Without sampling, testing one method on the XNLI dataset alone would take approximately 120 hours. Therefore, it has become a common practice, as demonstrated in previous works [1] [2] [3] [4] to evaluate LLM methods on a subset of the test set. It is important to note that the sampled test set is sufficiently large to avoid any potential bias.
>
>
>
> W1.2: Baseline
>
> Please refer to Response 3 in the Experiment part in the general response.
>
>
>
> W2: Low performance of LLMs on tasks
>
> The poor performance of Llama-7b can be attributed to its inability to effectively follow instructions. Additionally, in the case of XNLI, we have chosen to use labels such as "entailment", "contradict" and "neutral" instead of numerical values "0", "1", "2". This decision has made achieving an exact match more challenging. However, when we compare our results to those obtained using MAD-X and LF-LFT under the same conditions, our performance significantly improves, as demonstrated in the above first response.
>
>
>
> Q1: In Eq.3 and 4, how the symbol "~" is converted to "="?
>
> Similar to Eq.5 to Eq.6 and Eq.7 to Eq.8, there is hyperparameter $t$ in the merging method to balance the scale difference.
>
>
>
> Q2: relation between $\texttt{AdaMergeX}$ and MAD-X
>
> We have further explained the difference between $\texttt{AdaMergeX}$ and related works. Please refer to Response 1 and Response 2 in the Novelty part in the general response.
>
>
>
> [1] Pryzant et al, Automatic Prompt Optimization with "Gradient Descent" and Beam Search, Arxiv 2023.
>
> [2] Zhou et al, Large Language Models Are Human-Level Prompt Engineers, ICLR 2023.
>
> [3] Gonen et al, Demystifying Prompts in Language Models via Perplexity Estimation, EMNLP 2023.
>
> [4] We et al, Self-Adaptive In-Context Learning: An Information Compression Perspective for In-Context Example Selection and Ordering, ACL 2023.

---

> ### Author Response · Authors · 2023-11-22
> **Rquest more discussion with Reviewer tcUW**
>
> Dear Reviewer tcUW,
>
> Thanks so much for your valuable review, including (1) details of the training and testing dataset, (2) comparison with more cross-lingual transfer baselines, as well as (3) the relation between our methods and MAD-X. We have carefully responded to your concerns and conducted supplementary experiments to address your concerns. We are eagerly anticipating your feedback on our response and new experiments as the deadline for discussion draws near.
>
> Best regards,
> Authors

---

### Official Review · Reviewer_CTKw · 2023-11-01

**Soundness:** 3 good
**Presentation:** 3 good
**Contribution:** 2 fair
**Rating:** 5
**Confidence:** 5

**Summary:**

This work presents an approach to cross-lingual transfer where the idea is to merge adapters that can deal with 'general processing' of a source language and a target language (eliciting language abilities) further with the task adapter trained on the desired task in the source language (therefore eliciting 'task abilities'). The term 'structure-adaptive' adapter merging from the title and the abstract actually denotes the need to perform merging via different operations, which depends on the nature of the chosen adapter architecture: e.g., LoRA relies on elementwise addition, so the same operation should be used for adapter merging, while IA3 requires elementwise multiplication. As the ablations studies show, doing 'blind' merging that doesn't align with the actual underlying adapter structure yields large task performance drops.

The proposed merging strategy is then applied on several LLMs and mostly compared against other recent (and less sophisticated) merging strategies on several standard cross-lingual transfer tasks, spanning a total of 12 languages. The results show consistent gains over the chosen baselines.

**Strengths:**

While the paper is generally well written and easy to follow, for each of its strength there is a mirrored weakness. For each strength (S_i), I suggest to check the related weakness (labeled W_i later).

S1. The work connects the ideas of modular and PEFT learning (via adapters such as LoRA and IA3) on LLMs and cross-lingual transfer learning.

S2. One of the main conceptuals novelties, as claimed by the authors, is the division of information and abilities into 'language abilities' (captured through language adapters via causal language modeling) and 'task abilities' (captured through task-specific tuning). However, while it's interesting to revisit this idea in the context of LLMs, the idea is definitely not novel (see W2).

S3. The main results seem to suggest the gains of the proposed approach over the chosen baselines.

**Weaknesses:**

W1. The idea of connecting modular and PEFT learning with cross-lingual transfer has been explored before with encoder-only models (a body of work on bottleneck adapters, sparse subnetworks, etc.) as well as encoder-decoder models (e.g., check the work on mmT5). Moreover, even the idea of (simple) adapter merging is not novel and has been proposed, e.g., by Zhang+.

W2. There has been a large body of work that decomposed language and task abilities into dedicated language and task adapters and then performed various operations on such decomposed modules with well defined abilities. Cross-lingual transfer is basically one of the primary applications demonstrating how modularisation helps with postiive transfer. I suggest the readers to check a recent survey paper on modular deep learning of Pfeiffer+ for an overview (e.g., MAD-X work performed exactly this but stacking instead of merging language and task adapters). Overall, the paper doesn't perform a good job in contextualising their work within the wider area where the idea of modularisation for cross-lingual transfer has been extensively researched with encoder-only and encoder-decoder models. This diminishes the novelty of the work substantially.

W3. The gains are reported only over the chosen baselines (which seem most relevant at first), but there's a large body of work on cross-lingual transfer (i.e., with adapters as well as without adapters) that the paper simply ignores. For instance, combining language and task masks as done in the work of Ansell+ (ACL-22) can be seen as a form of direct adapter merging for cross-lingual transfer, and is therefore directly relevant as a baseline. Comparing performance to adapter-based transfer with encoder-only models is also a must, as previous work typically reported much higher absolute scores in general.

W4. The results in absolute terms are quite low - for instance, many XNLI results actually underperform the random baseline (or a majority baseline) in a 3-way classification task such as NLI. The same goes for XCOPA results. There are much higher scores reported on those benchmarks in prior work on cross-lingual transfer learning. Given the reduced novelty and other methods that are very relevant and perform some sort of adapter merging, I fail to see how exactly this approach advances the field.

W5. It is quite intuitive to see that merging adapters via the same technique that merges their parameters with the parameters of the original model will yield the highest performance. While it's nice to see this confirmed empirically, I feel that the paper overclaims this as a contribution (similar to overclaiming the novelty of decoupling learning into language and task adapters). Also, the work only explores adapters that get their parameters merged/composed with the original parameters of the large model, but it doesn't explore other techniques such as bottleneck and serial adapters, or a combination of different architectures (e.g., UniPELT or AutoPEFT).

W6 (Minor). The paper doesn't really evaluate on low-resource languages, but this mostly stems from the limitations of the underlying LLMs that simply cover less languages than models such as mT5, XLM-R, mDeBERTa, etc.

**Questions:**

Can the authors comment on low performance of LLMs on tasks such as XNLI and XCOPA that often go below the random/majority baseline?

Why haven't the authors compared the results also with encoder-based XLT approaches (e.g., MAD-X is one very relevant approach and there are also improved approaches that build on top of it)?

Have the authors also considered comparing to other adapter aggregation strategies in the context of XLT (e.g., adapting AdapterFusion for XLT)?

---

> ### Author Response · Authors · 2023-11-20
> **Response to CTKw**
>
> We thank the reviewer for your time and thorough response. We address the weaknesses and concerns of the reviewer below.
>
>
>
> W1 & W2: Novelty of "decompose language and task abilities"
>
> Please refer to Response 1 in the Novelty part in the general response.
>
>
>
> W5: Novelty of "adapter merging"
>
> Let us first response to the weakness 5 as it is also related to the novelty of our method. Please refer to Response 2 in the Novelty part in the general response.
>
>
>
> W3 & Q2: Other XLT baselines
>
> Please refer to Response 3 in the Experiment part in the general response.
>
>
>
> W4 & Q1: Low performance of LLMs on tasks
>
> The poor performance of Llama-7b can be attributed to its inability to effectively follow instructions. Additionally, in the case of XNLI, we have chosen to use labels such as "entailment", "contradict" and "neutral" instead of numerical values "0", "1", "2". This decision has made achieving an exact match more challenging. However, when we compare our results to those obtained using MAD-X and LF-LFT under the same conditions, our performance significantly improves, as demonstrated in the above first response.
>
>
>
> Q3: Other adapter aggression baselines
>
> In regards to AdapterFusion [1], AdapterSoup [2], LoRA-Hub [3], and other similar adapter aggregation approaches, they utilize a significantly larger amount of training data to obtain many adapter and encompass training parameters associated with fusion techniques. Hence, it would be unfair to draw comparisons between our method and theirs.
>
>
>
> [1] Pfeiffer et al, AdapterFusion: Non-Destructive Task Composition for Transfer Learning, EACL 2021.
>
> [2] Chronopoulou et al, AdapterSoup: Weight Averaging to Improve Generalization of Pretrained Language Models, EACL 2023.
>
> [3] Huang et al, LoraHub: Efficient Cross-Task Generalization via Dynamic LoRA Composition, Arxiv 2023.

---

> ### Author Response · Authors · 2023-11-22
> **Rquest more discussion with Reviewer CTKw**
>
> Dear Reviewer CTKw,
>
> We greatly appreciate your valuable comments on our work. We have carefully addressed your concerns regarding (1) the novelty of decomposing language ability and task ability; (2) more cross-lingual transfer baselines; (3) implementing our method on various adapters in the general response. As the deadline for discussion is approaching, we eagerly await your feedback on our response and new experiments.
>
> Thank you for your time and consideration.
>
> Best regards,
> Authors

---

> > ### Comment · Reviewer_CTKw · 2023-11-23
> >
> > Many thanks for providing the detailed response with further clarifications and additional experiments. The description of the main diff between what is considered 'language ability' and 'task ability' in this work versus prior work is now made much clearer to me, and the main contributions are easier to extract from the paper.
> >
> > I still do have concerns with the whole experimental setup, and I am not sure if the baselines such as MAD-X and SFT were properly optimised hparam-wise, so it would be good to see this in the paper as well.
> >
> > While the paper has definitely approved after the response, I still think there are other concerns not fully mitigated (e.g., mentioned by tcUW and PiPR) - I do increase my score though in appreciation of the response.

---

### Official Review · Reviewer_NmaX · 2023-11-01

**Soundness:** 4 excellent
**Presentation:** 4 excellent
**Contribution:** 3 good
**Rating:** 8
**Confidence:** 2

**Summary:**

This work introduces AdaMergeX, a new cross lingual transfer learning approach. The main idea is to realize that regular fine-tuning on a task consists of two aspects: “task ability”, the ability to train the actual task, and “language ability” the ability to understand the language in which the task was trained. With that assumption authors define a way to train a model on a specific task task and specific language by using another task as pivot to provide the language ability, to provide the task on a different language, and remove the undesired pair pivot task and languages.
This idea is adapted into two existing parameter efficient fine tuning methods, LoRA and (IA)3 and tested in a variety of tasks such as multilingual arithmetic reasoning, multilingual common-sense reasoning, multilingual natural language inference, question-answering and multilingual summarization. The method is compared against five cross-lingual transfer competing techniques, which are beaten by AdaMergeX.
A few ablation studies are presented, showing the generalizability of the approach regardless of the pivot language, using Spanish and Vietnamese instead, and generalizability  in terms of pivot task, comparing the performance using XNLI and XCOPA as reference tasks.
Finally an experiment using T5 instead of LlaMa model is performed, showing the generalizability  in terms of architecture.

**Strengths:**

The paper is sound and strong.
The idea is cleverly defined, implemented and tested.
The experimentation seems appropriate, it shows that the new approach is effective to perform cross lingual and cross task training.

**Weaknesses:**

Experimentation on this approach is difficult. All the experiments provide good evidence that support the author's claims, but given the generalizability nature of the approach, more experiments are needed.

**Questions:**

The ablation study on adaptive merging method is a bit confusing? What do the authors were expecting? Cross adaptive merging methods is probably a bad idea and your experiments support that.

The experiments on source language generalizability ? Why choose only 2 languages? Do the authors consider that the experiments on Spanish and Vietnamese make the point?

Similar question for XNLI and XCOPA for task generalizability

And for T5.

Do this approach work on Encoder models as well such as XLM or mBERT?

---

> ### Author Response · Authors · 2023-11-20
> **Response to NmaX**
>
> We thank the reviewer for your time and thorough comment. We address the weaknesses and concerns of the reviewer below.
>
> Q1: Ablation study on adaptive merging method
>
> In this ablation study, our aim is to demonstrate the crucial significance of the adaptive merging method. It emphasizes the necessity of applying different merging methods to corresponding adapters, as utilizing a cross-merging method would lead to a notable deterioration in performance.
>
>
> Q2: Generalizability on source languages
>
> Please refer to Response 1 in the Experiment part in the general response.
>
>
> Q3: Generalizability on reference tasks
>
> Please refer to Response 2 in the Experiment part in the general response.
>
>
> Q4: Generalizability on backbone models.
>
> Please refer to Response 3 in the Experiment part in the general response.

---

> ### Author Response · Authors · 2023-11-22
> **Response to NmaX**
>
> Dear Reviewer NmaX,
>
> Thank you very much for your appreciation of our work and valuable suggestions. We have seriously conducted more experiments to address your concerns. Specifically, we conducted experiments to verify the generalizability of our method corresponding to the source language, reference task, and backbone models. Please let us know if you have any other concerns. Thanks again for your time.
>
> Best regards,
> Authors

---

### Author Response · Authors · 2023-11-20
**General Response to Novelty**

1. **Novelty of "decompose language and task abilities"**

   Previous studies, such as MAD-X [1] and LF-SFT [2], have endeavored to untangle language ability from task ability. However, these studies operate under the assumption that task ability and language ability can be distinctively represented through individual adapters, disregarding their inherent interdependence. This assumption is fundamentally flawed and necessitates significant improvement. In particular, these studies derived task ability by fine-tuning on an English training set, thereby compromising the complete independence of the resulting task ability from language. Likewise, the acquisition of language ability in these studies relied on masked language modeling, which is also not a pure measure of language ability as it remains tied to the specific task.

   This will pose challenges when applied to other languages like French, Spanish, or low-resource languages such as Thai or Swahili. In such scenarios, MAD-X and LF-SFT would struggle to deliver satisfactory performance, as their effectiveness relies heavily on the specific characteristics of these source languages. Furthermore, when attempting to acquire language ability through training on more complex tasks like sentiment analysis, MAD-X and LF-SFT are bound to fail because they tangle language ability with task ability. [3] further supports this argument, proposing that "This deficiency might result in incompatibility between the Task Ability and the target Language Ability, which would emerge only at inference."

   On the contrary, we acknowledge that language plays a significant role in task ability and is connected to particular source languages, and vice versa. Hence, we define "language ability" as the disparity between the target language and the source language for each task. This disparity is calculated by assessing the divergence between adapters, employing the methodology described in Equation (4) of our proposed approach. In doing so, we effectively separate task ability and language ability into "task ability on the source language" and "language ability gap between the target language and the source language."

   We conduct following comparison to prove this point experimentally. We test on encoder-only multi-lingual models XLM-R and compare with MAD-X and LF-SFT. Results are listed here. In particular, we focus on the sentiment analysis task using the XNLI dataset as the reference task. Our findings reveal that when considering an encoder-only model, $\texttt{AdaMergeX}$ consistently outperforms MAD-X and LF-SFT in terms of performance.

   Results on XCOPA task:

   |           | tr   | vi   | th   | sw   | Avg. |
   | --------- | ---- | ---- | ---- | ---- | ---- |
   | MAD-X     | 60.3 | 66.1 | 61.8 | 56.3 | 59.5 |
   | AdaMergeX | 69.4 | 70.5 | 66.9 | 63.2 | 67.5 |

   Results on XQuAD task:

   |           | el   | ru   | th   | tr   | Avg. |
   | --------- | ---- | ---- | ---- | ---- | ---- |
   | MAD-X     | 54.3 | 57.8 | 55.7 | 51.1 | 54.7 |
   | LF-SFT    | 65.5 | 64.6 | 75.2 | 58.6 | 66.0 |
   | AdaMergeX | 70.2 | 70.4 | 77.9 | 63.8 | 70.6 |

2. **Novelty of "adaptive adapter merging"**

   Firstly, in comparison to the original simple addition merging method, adaptive adapter merging showcases a substantial improvement in performance, as shown in Table 2.

   Secondly, for more kind of adapters, we claim in the paper that "In the case of AdapterH and Prefix-Tuning, which involve the insertion of layers and prefix tokens into the model, the merging process necessitates the usage of MLP to transfer adapters within the same space. In this paper, we primarily focus on conducting LoRA and (IA)$^3$ due to the lack of enough training data for adapterH and the subpar performance of prefix-tuning on fine-tuning."

   However, despite the lack of sufficient training data for Llama2-7b, there is an adequate amount of training data available for smaller models like mT5-base. Consequently, we have chosen to implement our method using prefix tuning [4] with mT5-base. Formally, the adaptive merging method is $A_{l_1t_1} = t \cdot (A_{l_1t_2} * A_{l_2t_2}^{-1}) * A_{l_1t_1}$, where $*$ represents matrix multiplication and $A_{l_2t_2}^{-1}$ represents Moore-Penrose pseudo-inverse of the matrix. The results demonstrate that $\texttt{AdaMergeX}$ excels remarkably within the realm of prefix-tuning, a distinct and separate approach to fine-tuning.

   Results on XNLI task:

   | Method    | es   | fr   |
   | --------- | ---- | ---- |
   | Eng-FT    | 31.2 | 29.7 |
   | AriMerge  | 29.8 | 28.3 |
   | AdaMergeX | 34.1 | 31.4 |


[1] Pfeiffer et al, MAD-X: An Adapter-Based Framework for Multi-Task Cross-Lingual Transfer, EMNLP 2020.

[2] Ansell et al, Composable Sparse Fine-Tuning for Cross-Lingual Transfer, ACL 2022.

[3] Marinela et al, Cross-Lingual Transfer with Target Language-Ready Task Adapters, ACL 2023.

---

### Author Response · Authors · 2023-11-20
**General Response to Experiment**

1. **More ablation analysis on source language.**

   We have added more experiments to support the generalizability of our approach. We test on five source languages including German, French, Spanish, Thai, Vietnamese. Results are listed here, and the dataset was tested on the available languages, including German, French, Spanish, Thai, and Vietnamese. We consistently find that $\texttt{AdaMergeX}$ performs well across various source languages.

   Results with LoRA:

   | Source Language | Method           | MGSM | XCOPA | XNLI | XQUAD | Avg. |
   | --------------- | ---------------- | ---- | ----- | ---- | ----- | ---- |
   | German          | German-Tuned     | 20.9 | -     | 48.3 | 44.4  | 37.9 |
   |                 | AdaMergeX        | 22.3 | -     | 50.9 | 46.5  | 39.9 |
   | French          | French-Tuned     | 19.9 | -     | 52.9 | -     | 36.4 |
   |                 | AdaMergeX        | 22.2 | -     | 57.1 | -     | 39.6 |
   | Spanish         | Spanish-Tuned    | 19.2 | -     | 33.9 | 45.4  | 32.8 |
   |                 | AdaMergeX        | 18.7 | -     | 35.1 | 49.1  | 34.3 |
   | Thai            | Thai-Tuned       | 3.2  | 49.3  | 1.9  | 39.8  | 23.6 |
   |                 | AdaMergeX        | 4.5  | 48.9  | 6.2  | 44.2  | 26.0 |
   | Vietnamese      | Vietnamese-Tuned | -    | 63.8  | 49.1 | 36.2  | 49.7 |
   |                 | AdaMergeX        | -    | 64.2  | 53.2 | 38.9  | 52.1 |

   Results with IA$^3$

   | Source Language | Method           | MGSM | XCOPA | XNLI | XQUAD | Avg. |
   | --------------- | ---------------- | ---- | ----- | ---- | ----- | ---- |
   | German          | German-Tuned     | 2.9  | -     | 43.5 | 45.6  | 30.7 |
   |                 | AdaMergeX        | 6.3  | -     | 44.0 | 47.1  | 32.5 |
   | French          | French-Tuned     | 2.5  | -     | 48.7 | -     | 25.6 |
   |                 | AdaMergeX        | 4.1  | -     | 47.9 | -     | 26.0 |
   | Spanish         | Spanish-Tuned    | 3.5  | -     | 49.2 | 45.9  | 32.9 |
   |                 | AdaMergeX        | 5.3  | -     | 50.9 | 44.6  | 33.6 |
   | Thai            | Thai-Tuned       | 1.2  | 49.8  | 0    | 27.7  | 19.7 |
   |                 | AdaMergeX        | 1.9  | 50.4  | 0    | 28.9  | 20.3 |
   | Vietnamese      | Vietnamese-Tuned | -    | 49.8  | 45.5 | 33.2  | 42.8 |
   |                 | AdaMergeX        | -    | 48.7  | 50.2 | 36.1  | 45.0 |

2. **More ablation analysis on reference task.**

   We test on three different reference tasks, including XCOPA, XNLI, XQuAD. Results are listed here, and the dataset was tested on the corresponding available languages among German, French, Spanish, Thai, and Vietnamese. We consistently find that AdaMergeX performs well across various reference tasks.

   Results with LoRA:

   | Reference Task | Method        | MGSM | XCOPA | XNLI | XQUAD | Avg. |
   | -------------- | ------------- | ---- | ----- | ---- | ----- | ---- |
   |                | English-Tuned | 14.4 | 59.9  | 44.6 | 42.3  | 40.3 |
   | XCOPA          | AdaMergeX     | 15.2 | 60.2  | 45.1 | 43.8  | 41.1 |
   | XNLI           | AdaMergeX     | 14.5 | 60.9  | 46.7 | 44.1  | 41.6 |
   | XQuAD          | AdaMergeX     | 14.9 | 61.8  | 45.4 | 44.4  | 41.6 |

   Results with IA$^3$ :

   | Reference Task | Method        | MGSM | XCOPA | XNLI | XQUAD | Avg. |
   | -------------- | ------------- | ---- | ----- | ---- | ----- | ---- |
   |                | English-Tuned | 2.6  | 52.7  | 40.0 | 39.2  | 33.6 |
   | XCOPA          | AdaMergeX     | 4.9  | 54.3  | 40.5 | 40.4  | 35.0 |
   | XNLI           | AdaMergeX     | 3.6  | 54.6  | 41.2 | 39.9  | 34.8 |
   | XQuAD          | AdaMergeX     | 4.1  | 53.9  | 42.1 | 41.0  | 35.3 |

3. **More backbone models and Cross-Lingual Transfer Baselines**

   We test on encoder-only multi-lingual models XLM-R and compare with MAD-X and LF-SFT. Results are listed here. In particular, we focus on the sentiment analysis task using the XNLI dataset as the reference task. Our findings reveal that when considering an encoder-only model, AdaMergeX consistently outperforms MAD-X and LF-SFT in terms of performance.

   Results on XCOPA task:

   |           | tr   | vi   | th   | sw   | Avg. |
   | --------- | ---- | ---- | ---- | ---- | ---- |
   | MAD-X     | 60.3 | 66.1 | 61.8 | 56.3 | 59.5 |
   | AdaMergeX | 69.4 | 70.5 | 66.9 | 63.2 | 67.5 |

   Results on XQuAD task:

   |           | el   | ru   | th   | tr   | Avg. |
   | --------- | ---- | ---- | ---- | ---- | ---- |
   | MAD-X     | 54.3 | 57.8 | 55.7 | 51.1 | 54.7 |
   | LF-SFT    | 65.5 | 64.6 | 75.2 | 58.6 | 66.0 |
   | AdaMergeX | 70.2 | 70.4 | 77.9 | 63.8 | 70.6 |

---

### Author Response · Authors · 2023-11-21
**General Reply**

We have updated a new PDF version, which includes several significant updates. The key changes are as follows:
1. Updates Figure 1: We have made enhancements to Figure 1 and revised the related content in Introduction and Section 3.1 to better illustrate the novelty of $\texttt{AdaMergeX}$ when decoupling task ability and language ability.
2. Expanded Experiment on More Kinds of Adapters: Section 3.2 now includes an expanded experiment that explores the merging of more kinds of adapters. This illustrates the novelty of $\texttt{AdaMergeX}$ on adapter merging.
3. Expanded Experiment Results in Section 4.3: In response to valuable feedback, we have included additional baselines and conducted more comprehensive ablation analyses in our updated experiment results.
4. Addressing Reviewer Feedback: We have taken into account the writing issues raised by the reviewers and made the necessary revisions to ensure clarity and coherence.

We greatly appreciate the valuable input you have provided.

---

### Meta-Review · Area_Chair_51iF · 2023-12-08

**Metareview:**

This paper proposes AdaMergeX to improve cross-lingual transfer by merging language-specific generic-task adapters with task-specific adapters in a different language. To make the merging operation work, a new merging method is proposed that differs depending on whether LoRA or IA3-based adapters are being merged. The merging operation for LoRA adapters resembles past work, but the IA3 merging operation involves division and multiplication to reflect the structure of the IA3 updates (which represents the main novelty of this work). The proposed framework has strong performance, but reviewers were all generally concerned about this work not situating itself appropriately in the large literature on modular models and cross-lingual generalization. While the authors did include some additional baseline methods and discussion during the rebuttal, the scale of the issues with situating the work necessitates substantial more revision and resubmission.

**Justification For Why Not Higher Score:**

As discussed in the metareview, the paper is missing a large body of work that should be compared and discussed.

**Justification For Why Not Lower Score:**

N/A

---

### Decision · Program_Chairs · 2024-01-16

Reject